# Factors Associated with Vaccination Coverage among 0–59-Month-Old Children: A Multilevel Analysis of the 2020 Somaliland Demographic and Health Survey

**DOI:** 10.3390/vaccines12050509

**Published:** 2024-05-08

**Authors:** Mohamed Abdalle Osman, Alexander Waits, Li-Yin Chien

**Affiliations:** 1International Health Program, Yang-Ming Campus, National Yang Ming Chiao Tung University, Taipei City 112304, Taiwan; baallesamte@gmail.com; 2Faculty of Health Sciences, Sanaag University, Erigavo, Somaliland; 3Institute of Public Health, Yang-Ming Campus, National Yang Ming Chiao Tung University, Taipei City 112304, Taiwan; alexwaits@nycu.edu.tw; 4Institute of Community Health Care, College of Nursing, Yang-Ming Campus, National Yang Ming Chiao Tung University, Taipei City 112304, Taiwan

**Keywords:** vaccination, vaccination coverage, child, Somaliland, multilevel analysis, Demographic Health Survey

## Abstract

Globally, there has been little growth in vaccination coverage, with countries in the Horn of Africa having the lowest vaccination rates. This study investigated factors associated with vaccination status among children under five years old in Somaliland. The 2020 Somaliland Demographic and Health Survey surveyed women aged 15–49 years from randomly selected households. This multilevel analysis included 2673 primary caregivers of children under five. Only 34% of children were ever vaccinated. Childhood vaccination coverage was positively associated with high-budget regions, high healthcare facility density, and children older than 23 months. Vaccination coverage was greater for urban and rural residents than for nomadic people. Children whose mothers could read part of one sentence or one complete sentence were more likely to be vaccinated than illiterate mothers. Children whose mothers received antenatal care (ANC) once, two to three times, or four times or more were more likely to be vaccinated than those whose mothers received no ANC. Childhood vaccination coverage in Somaliland is low. Promoting maternal ANC visits and increasing women’s literacy may enhance vaccination coverage. Funds should be allocated to areas with low resources, particularly for nomadic people, to boost vaccination uptake.

## 1. Introduction

Child vaccination is the most cost-effective public health intervention for reducing neonatal and child mortality [1]. The World Health Organization (WHO) proposed a global childhood immunization coverage target of 90% to achieve a neonatal mortality of less than 12 per 1000 live births and child (younger than five) mortality rates lower than 25 per 1000 live births by 2030 [2]. Approximately three to five million premature deaths among children under five years of age are prevented by childhood immunization worldwide [3,4,5]. To expand childhood immunization programs, eliminate vaccine-preventable infectious diseases, and distribute vaccinations among children efficiently and universally, the WHO initiated an expanded program of immunization (EPI) in 1974 [6]. Since the initiation of the EPI program, a 29% reduction in five-year child mortality has been achieved, where an estimated 17% of under-five deaths were caused by diseases that could have been prevented by vaccinations [5,7,8,9].

The global vaccination coverage among children aged 0–59 months was estimated to be 81% in 2020. Because of the COVID-19 pandemic, global vaccination coverage among children under five years of age decreased by 5%, compared to 86% in 2018 [10]. This coverage rate (81%) is similar to the rate in 2010, showing a significant decrease in childhood vaccination coverage during the pandemic [11]. Globally, more than 22.7 million children do not complete essential vaccinations, accounting for 17% of children under five years of age [12]. For instance, 17.1 million children had not received any dose of the diphtheria, tetanus, and pertussis vaccine [12]. Low- and middle-income countries are reported to have low childhood vaccination coverage [13,14,15]. Notably, sub-Saharan African (SSA) countries have the lowest childhood vaccination coverage rate. The pooled prevalence in SSA countries was 69.2%; Ethiopia had the lowest coverage rate (39.5%) in the Horn of Africa, while Somalia and Somaliland were not included [16].

In Somaliland, the EPI launched in 1995 with WHO and UNICEF backing. It began in cities and expanded nationally, following WHO immunization guidelines. This includes doses for the BCG, pentavalent, polio, and measles vaccines. Somaliland now has the world’s second-lowest vaccination coverage for under-fives, while Somalia has the lowest at 11% [17]. In Somaliland, only 13% of children under two are fully vaccinated, with 67% never immunized [18]. Additionally, the under-five mortality rate, at 91 deaths per 1000 live births, is the highest in East Africa, except for Somalia [19,20].

Somaliland, a low-income country in the Horn of Africa has a population of 4.2 million, with 37.8% under 15 years of age and 20% nomadic. It declared independence from Somalia in 1991 after a deadly war, rebuilding its healthcare system from scratch. Yet, due to economic and infrastructure challenges, it continues to reconstruct its primary healthcare system, vital for universal health coverage [21].

Most studies on vaccination coverage for children under five years old found maternal education, wealth, birthplace, residence, perinatal care access, family income, size, and childbirth order are determinants of vaccination uptake [13,22,23,24,25,26]. Notably, a cross-sectional study conducted in Yemen emphasized that paternal low education and maternal young age at first birth were negatively associated with vaccination coverage among children under five years of age [27]. In particular, studies conducted in SSA countries have consistently revealed that poor households, uneducated parents, mothers’ low exposure to media, and illiteracy substantially reduced the odds of children being immunized [16,28]. Furthermore, a cross-sectional study conducted in the Somali region in Ethiopia showed a gender disparity, where being a male-sex child was positively associated with childhood vaccination uptake [29]. Previous studies often used high thresholds for variables, which may not fit Somaliland’s context. They focused on mothers with four or more ANC visits and formal education, underestimating those with one or two visits or no formal education but with some reading abilities. Additionally, urbanicity, community literacy, and women’s education consistently predict higher childhood immunization rates [16,30,31]. However, the analysis did not include nomadic people, who constitute more than 20% of Somaliland’s population, and who often migrate with livestock, facing socioeconomic and healthcare access challenges.

Other studies highlighted that regions of residence and communities with greater utilization of ANC services were positively associated with childhood vaccination coverage [32,33]. Moreover, a systematic review in India highlighted that infants from communities with better access to healthcare services have better vaccination coverage [34]. However, these studies overlooked regional differences in healthcare facility density and unequal budget allocation, both potentially impacting outcomes. Regions with more resources likely have better healthcare infrastructure and access than those with fewer resources.

Although the literature in Somaliland is limited, a multiple indicator cluster survey in 2011 and a recently published cross-sectional study reported that lack of maternal awareness, living in rural and nomadic areas, unemployed mothers, and low maternal education were associated with low participation in the child immunization campaign as well as low uptake of child vaccinations [35,36]. Despite Somaliland’s low childhood vaccination coverage and high childhood mortality rates, there’s a notable scarcity of nationally representative studies examining the individual and contextual factors linked to childhood vaccination coverage. This study sought to fill this gap by analyzing the individual and regional factors influencing vaccination coverage among children under five years old in Somaliland. Vaccine hesitancy and acceptance are intricately tied to socio-cultural dynamics, underscoring the importance of contextualizing and comparing factors across different regions. Notably, to our knowledge, this is the first multilevel analysis focusing on childhood immunization coverage conducted in Somaliland.

## 2. Materials and Methods

### 2.1. Study Settings, Data Sources, and Design

This study is a secondary analysis of the data from the Somaliland Demographic and Health Survey 2020 (SLDHS 2020). The SLDHS2020 is the first countrywide household survey that collected demographic and health data in all six main regions of Somaliland. The survey participants were women aged 15–49 years. The sampling design used a multistage probability sampling method, where each region was stratified into urban, rural, and nomadic residential regions. Based on the six regions and their three residential urbanicity levels (urban, rural, and nomadic), 18 equal strata were finally generated. In the first stage, 35 primary sampling units (PSUs) were randomly selected for each stratum in the urban and rural areas. Ten PSUs were randomly selected from nomadic areas. In the second stage, 10 secondary sampling units (SSUs) were randomly selected from 35 listed PSUs in the urban and rural strata only. In the last stage, 30 households were randomly selected from each assigned PSU, and women of reproductive age (15–49) in the selected households were eligible to participate. Information about maternal and child (under five years of age) health was collected from all women. Trained interviewers collected the data, and the interviews were conducted face-to-face using validated questionnaires (SLDHS 2020).

### 2.2. Study Participants

Verbal consent was obtained from participants due to low literacy rates in most communities. The consent letter was read aloud, and participants were informed that their names and personal identification would be completely anonymous. They were also informed of their right to refuse and stop the interview at any time. The response rate was 92.8%. This secondary data analysis included all mothers or primary caregivers of children aged 0–59 months (n = 2987). However, 314 (10%) of the participants did not have data regarding childhood vaccinations and thus were excluded. The final analytical sample size was 2673. This study has been approved by the National Ethical Committee along with the Unit of Research Development, Ministry of Health Development of Somaliland (MOHD/DG:2/296/2023). We accessed the SLDH 2020 data for research purposes on 18 June 2022. We did not have access to information that could identify individual participants. The data are available given permission from Central Statistics, the Department of Ministry of National Planning, Somaliland. 

### 2.3. Measurements

#### 2.3.1. Dependent Variables 

The outcome variable for children’s vaccination status was measured using vaccination information in childhood vaccination cards (4%) and the mother’s or primary caregiver’s recall (96%). If the child’s vaccination card was available, the information regarding vaccination status was recorded from the card; if not, the mothers or primary caregivers were asked if their child had ever received any shot of childhood vaccinations. Subsequently, based on the response, the outcome was categorized into never vaccinated (not having received any childhood vaccine) and ever vaccinated (having received at least one childhood vaccine). Information about specific vaccines was not sufficiently available because the participants could not recognize the names of the specific vaccines. The proportions with missing data were 61.4%, 67.1%, 71.5%, and 43.5% for the BCG, polio, pentavalent, and measles vaccines, respectively. Therefore, we created a variable based on whether the child ever received any childhood vaccine instead of a specific vaccine.

#### 2.3.2. Independent Variables

Individual-level factors included parental or primary caregiver sociodemographic factors, child-related factors, and maternal healthcare utilization. The respondents’ ages were grouped into ≤24 years, 25 to 34 years, and ≥35 years. Maternal education was measured by asking if they ever attended school and which level of education they achieved, such as primary, secondary, or above. Maternal literacy was measured by asking participants if they could read one sentence in the local language; if participants could not read, they were categorized as illiterate. Maternal media exposure was measured by “how often the respondents listen to the radio, watch television, or read magazines per week”. If they listened, watched, or read one of the media channels at least once a week, they were exposed; if not, they were considered not exposed. The respondents’ marital status was categorized as married, divorced, or widowed. Parental employment status was dichotomized as working (employed) or not working (unemployed). Mothers’ parity was categorized into three categories: 1 to 2, 3 to 5, and ≥6. The wealth index was categorized as poor, middle, or rich. The wealth index was derived from household characteristics; the ownership of the house and land; the presence of a car, motorcycle, or bicycle; and the availability of electricity, television, and refrigerators in the home. Five wealth quintiles were initially generated; the first two wealth quintiles were grouped as poor, the third and fourth quintiles were grouped as middle, and the fifth quintile as rich. The number of children at home was categorized into three groups (1 to 2, 3 to 4, and ≥5).

Maternal healthcare utilization was measured by the mother’s number of ANC visits. Maternal ANC visits were classified into four categories (no visits, one visit, two to three visits, and four or more visits). Child-related factors included sex, age, birth order, place of birth, and place of residence. The mothers were asked about the age of their youngest child in months, and the children were grouped as ≤23 months old or 24–59 months old. The place of delivery was dichotomized (at a health facility or home). Residence status was categorized as urban, rural, or nomadic. Birth order was classified as first, second, third, or later.

The regional-level variables were based on the six main regions of the SLDHS. Budget allocation and the density of healthcare facilities in the six regions were extracted and derived from reports released by the Ministry of Finance and the Central Statistics Department. The percentage of national budget allocation per region was found in the national budget report published by the Ministry of Finance Development [37]. The number of healthcare facilities, including public hospitals, mother and child health centers, health posts, and mobile health teams, was aggregated, and the density of public healthcare facilities across the region was generated [38,39]. Based on the proportion of public healthcare facilities distributed across the regions, we used the median as the cutoff point, and those below the median were considered regions with low healthcare facility density. We did the same for the national budget allocation regions, where regions that received percentages below the median level of the national budget allocated by the central government were considered low-budgeting regions.

#### 2.3.3. Data Analysis

The data were analyzed using Stata version 15 (Stata statistical software: release 15. College Station, TX, USA: Stata Corp.). Cross-tables of ever-vaccinated status according to respondent’s sociodemographic characteristics, children, and region-level factors were analyzed using X2 statistics. Since the SLDHS-2020 is a nationwide survey, both bivariate and multilevel multivariate analyses were weighted using the women’s sample weights in the SLDHS dataset. Additionally, the SLDHS 2020 was designed in a hierarchical structure, where individual factors were nested within the clusters or community, which may cause children belonging to the same cluster to have similar characteristics to those belonging to other clusters and, hence, may violate the assumption of the binary logistic regression model. Therefore, multilevel logistic regression of a hierarchical random effects model was used to identify the associations of individual- and regional-level factors with childhood vaccination status.

The intraclass correlation coefficient (ICC) and proportional change in variance (PCV) were employed to measure the variation between clusters in the random effects model. Four models (model I, model II, model III, and model IV) were built. The null model (model 1) was used to test the community variability in childhood vaccinations between clusters and to determine whether the data fit the multilevel modeling. The second model (Model II) employed individual-level factors, the third model (Model III) included regional-level factors only, and the fourth model (Model IV) included both individual- and regional-level factors. Finally, the parsimonious model (model V) was employed, and only those variables that significantly predicted the outcome were retained in this model. The parsimonious model was considered the final model of this analysis since it showed sufficient ICCs and good model fit. Adjusted odds ratios (AORs) and 95% confidence intervals (CIs) are reported. A *p*-value < 0.05 was considered to indicate significance. The log-likelihood ratio (LR) and Akaike information criterion (AIC) were used to check the model fitness. Multicollinearity was checked using the variance inflation factor (VIF) whenever a VIF < 10 and tolerance greater than 0.1 indicated the absence of multicollinearity.

## 3. Results

The characteristics of the study participants are presented in Table 1. Of the 2673 children included, 910 (34%) were found to have received at least one dose of childhood vaccine. A report of individual vaccines is presented in the Appendix A. Almost half of the respondents (51.3%) were in the 25- to 34-year-old age group, 2253 (84.3%) never attended school, and 2142 (80%) were illiterate. More than two-thirds (69.9%) of the mothers had a poor wealth index, while nearly all of the mothers (95.2%) were married and not working (95.5%). Over four-fifths (86%) of mothers did not receive ANC, and 85% delivered at home. Approximately two-thirds (63%) of the respondents were from low-budgeting regions, and 84.7% were from low healthcare facility density regions. Close to two-thirds (64.7%) of the fathers were unemployed. Almost half (51.7%) of the children were male, 26% were first-born babies, and approximately half (52%) had nomadic residences.

The characteristics of the participants by vaccination coverage status are presented in Table 1. Among children ever vaccinated, 22.6% had mothers ≤ 24 years of age, compared to 30.2% among those who were not vaccinated. Among children ever vaccinated, 27.6% were born to mothers with some level of education, compared to 12% among those who were never vaccinated. Among children who were ever vaccinated, 64% had illiterate mothers. Among children who were never vaccinated, 85% had illiterate mothers.

Additionally, among children who were ever vaccinated, 34.8% were from low-income families, compared to 67.2% among those who were never vaccinated. Among children who were ever vaccinated, 55% had working fathers, compared to 25.4% among unvaccinated children. Among children ever vaccinated, 11% had mothers who had ≥4 ANC visits, compared to 3.5% among those who were unvaccinated. Furthermore, among those who were ever vaccinated, 68.3% were older than 23 months of age, compared to 47.4% among those who were not vaccinated. Among those who were ever vaccinated, 77.2% were born first or second, compared to 68.2% among those who were never vaccinated.

In addition, among children who were ever vaccinated, 22% were delivered at health facilities, compared to 11.5% among those who were not vaccinated. Furthermore, among those who were ever vaccinated, 75.6% had mothers who had no exposure to the media, compared to 89.7% among those who were not vaccinated. Among those who were ever vaccinated, 50% lived in urban areas, compared to 19% among those who were not vaccinated.

For regional variables, among those ever vaccinated, 72.8% were from low healthcare facility density areas, compared to 85.3% among those not vaccinated. Among those who were ever vaccinated, 45.2% were from low-budgeting regions, compared to 63.5% among those who were not vaccinated. All these differences were significant. 

Table 2 presents the multilevel logistic regression analysis of individual- and community-level factors associated with childhood vaccination coverage among children under five years of age in Somaliland. In this multivariate-multilevel analysis, five models were employed (see Section 2.3.3). 

Random effect analysis showed variation in childhood vaccinations across the clusters, where almost 29% (ICC = 0.288 in Model I) of childhood vaccination variability was due to cluster differences in the null model. The parsimonious model (Model V) revealed a PCV of 0.66, indicating that 66% of the variation in childhood vaccination at the community level can be attributed to combinations of individual- and community-level factors. According to the model fitness, the parsimonious model (Model V) had a much lower AIC (AIC = 2974.877) than the null model (AIC = 3179.903), which showed good model fitness. Furthermore, the parsimonious model (Model V) had a greater ICC (9.9%) than the full model (9.0%) and was selected as the final model (Table 2).

The final parsimonious model included only those variables significantly associated with the outcome in Model four (the full model). When comparing the estimates of all models, this study’s conclusions and main findings are based on the parsimonious model. The findings of the final model indicated that maternal literacy status, maternal number of ANC visits, age of the children, place of residence, healthcare facility density, and national budget allocation were significantly positively associated with primary childhood vaccination status. The results suggested that children whose mothers could read at least part of one sentence (AOR = 1.61, 95% CI: 1.07–2.41) or one complete sentence (AOR = 1.89, 95% CI: 1.27–2.80) had a greater chance of receiving childhood vaccinations than those whose mothers were illiterate. Children whose mothers received ANC once (AOR = 2.06, 95% CI: 1.31–3.24), two to three times (AOR = 1.97, 95% CI: 1.34–2.90), or four or more times (AOR = 2.08, 95% CI: 1.29–3.35) were more likely to be vaccinated than those whose mothers did not receive any ANC. Children older than 23 months had 2.02 (AOR = 2.02, 95% CI: 1.58–2.58) greater odds of being ever vaccinated than those younger than 23 months. Children from urban areas were 7.45 (AOR = 7.45, 95% CI: 4.54–12.2) times more likely to be ever vaccinated than those from nomadic areas. Similarly, children living in rural areas were 4.86 (AOR = 4.86, 95% CI: 2.30–10.2) times more likely to be ever vaccinated than those living in nomadic areas. Children from regions with a high density of healthcare facilities were more likely to be vaccinated (AOR = 1.64, 95% CI: 1.05–2.57) than those from the regions with a low density of healthcare facilities. Similarly, children in regions with high budgets had 1.62 (AOR = 1.62, 95% CI: 1.11–2.36) greater odds of being vaccinated than those in regions with low budgets (Table 2).

## 4. Discussion

This study aimed to investigate the individual- and regional-level factors associated with childhood vaccinations among children aged 0 to 59 months in Somaliland using the SLDHS-2020. The results revealed that only 34% of children under five years old had received any childhood vaccinations. The coverage surpasses neighboring Somalia, where only 11% of children of similar ages are vaccinated [40]. This difference can be attributed to the aftermath of civil war disruptions in the early 1990s. Somaliland has enjoyed political stability and peace for the past three decades and focused on rebuilding security, infrastructure, and healthcare systems, including the EPI program. In contrast, Somalia’s efforts in these areas have been hampered by ongoing instability. However, this coverage is much lower than that of other East African countries, such as Kenya (77%), Uganda (57.4%), and Rwanda (56.1%) [16,41]. Poor vaccination coverage is an important public health concern because young people remain unprotected from vaccine-preventable diseases. Therefore, young children in Somaliland need urgent attention from policymakers to address the barriers to receiving vaccines.

Only 108 (4%) of the 2673 children under five years of age included in this study had vaccination cards or documents, as presented in the Appendix A, which is extremely low compared to Ethiopia (the closest neighboring country), where a study reported that 74.5% of children had vaccination cards [42]. This very low prevalence of having vaccination documents requires stakeholders and policymakers to prioritize and attend to the issue. Making health cards or vaccination documents compulsory for children enrolling in preschools (kindergarten) or schools may improve childhood vaccination coverage in Somaliland.

This study revealed a positive association between maternal literacy and childhood vaccination coverage. Children whose mothers could read at least one sentence were more likely to be ever vaccinated than were those whose mothers were illiterate. Since the Somaliland population has a low literacy rate, the SLDHS evaluated literacy by assessing whether the participant could read the part or the whole of one sentence as a cutoff point for literacy status. This finding is consistent with other studies conducted in Ethiopia, the DRC-Congo, India, Bangladesh, and Nepal, showing that maternal education plays a significant role in childhood immunization uptake. However, these studies used education level to measure women’s literacy status [15,33,43,44,45]. The association between literacy status and vaccination status may be explained by the fact that educated mothers and those with higher literacy may be more aware of the importance of immunization. Likewise, literate mothers can read written posters and acquire information about the importance of the childhood vaccination schedule. Maternal literacy plays a role in children’s health, including in the context of immunization. The government should invest in women’s education, such as informal adult education, which can increase women’s literacy in Somaliland. Interestingly, even being able to read a part of one sentence was significantly associated with childhood vaccinations, which implies that even minimal investments in women’s education and literacy can significantly improve childhood immunization coverage.

Maternal ANC visits were positively associated with childhood vaccination uptake. Mothers who had at least one ANC visit were twice as likely to vaccinate their children than those with no ANC visit. This finding is consistent with other literature indicating the importance of maternal ANC utilization in childhood vaccination [46,47,48,49]. Although four ANC visits are considered standard in Somaliland, we observed that having only one ANC visit increased the odds of their children being vaccinated compared to children of mothers who did not visit ANC at all. In Somaliland, counseling is provided to each expecting mother during her ANC visits to promote awareness of ways to improve health and the importance of immunization to protect herself and her children. Moreover, a dose–response relationship between maternal ANC visits and outcomes was expected, but all three categories (1, 2–3, and 4+ ANC visits) had almost the same odds of having vaccinated children as those who did not receive ANC. A possible explanation for this can be that due to the low ANC visits and poor adherence to perinatal care services among women in Somaliland, healthcare service providers tried to provide all required medical services, including consultation, screening, immunization counseling, and any medical service that they could for the first visit and any subsequent visits, regardless of whether they made a specific appointment for a particular service. In particular, maternal and child health staff in rural settings are aware of accessibility constraints and low community awareness, so mothers are not guaranteed to return to health centers. Hence, interventions promoting maternal health literacy, such as ANC utilization, may improve childhood immunization coverage in Somaliland. Additionally, increasing community awareness about the importance of childhood immunization and maternal ANC utilization may increase childhood vaccination uptake.

The results also revealed a significant association between place of residence and childhood vaccination status. Children living in urban and rural areas are more likely to be vaccinated than those living in nomadic areas. This result is supported by previous studies conducted in SSA regions such as Nigeria, Ghana, Ethiopia, Burkina Faso, and Kenya, which found that urbanicity positively predicts childhood immunization coverage [50,51,52,53,54,55]. Access to healthcare facilities and informative broadcasting networks is more available to urban residents, which creates more childhood vaccination opportunities. Notably, Somaliland has a high percentage of nomadic pastoralists, while most other countries have only urban and rural residents. These nomadic people do not have permanent dwellings; they are mostly temporary inhabitants who move from area to area, which limits opportunities to receive basic vaccinations at an early age. The government might consider strategies to improve vaccination delivery among nomadic communities during the planning and implementation of the national EPI. Establishing specialized mobile teams that can move with nomadic people may improve the effectiveness of the EPI in nomadic communities. Training and empowering the nomadic community by recruiting community leaders and religious leaders in immunization campaigns may improve community trust and adherence to immunization services.

Our analysis revealed that national budget allocation significantly predicts childhood vaccination uptake, with children in high-budget regions more likely to be vaccinated than those in low-budget regions, aligning with prior research findings [16]. The substantial variation in budget distribution among regions notably impacted vaccination coverage, emphasizing the need for fair budget distribution to eliminate disparities. Similarly, healthcare facility density across regions significantly influenced vaccination coverage, with low-density areas experiencing lower coverage. Consistent with previous research, community-level factors such as budget allocation and healthcare facility density positively predicted childhood vaccination, suggesting that allocating resources and infrastructure to low-density regions can enhance coverage and reduce disparities [56,57,58,59,60]. Possible justifications suggest that high-budget regions have the financial resources to finance healthcare delivery and infrastructure and expand primary healthcare coverage, such as childhood vaccination campaigns.

In some rural and nomadic areas, the feasibility of immunization is limited due to financial constraints, inadequate infrastructure, and insufficient energy resources, exacerbated by the scarcity of health centers and electricity supplies. The availability of vaccines is not a problem in urban and rural areas with maternal and child health centers, while appropriate storage of vaccines is a problem in nomadic areas. Increasing vaccination availability is a major challenge in rural and nomadic settings without maternal and child health centers. Rural and nomadic settings often do not have a stable electricity supply. Since vaccines need to be kept at certain temperatures, the government implemented a national immunization campaign one to three times a year, depending on vaccine type. For example, a measles vaccination campaign is held once per year, whereas polio and pentavalent vaccine campaigns are held three times per year. However, these campaigns are insufficient due to the high fertility rate of the nomadic population and their irregular movement across the country. Currently, less than 3% of the national budget is allocated to the healthcare system. Therefore, additional investment should be directed toward the healthcare sector of Somaliland. The distribution of vaccines should adopt door-to-door, house-to-house, and family-to-family approaches during childhood vaccination campaigns.

Finally, the bivariate analysis revealed that maternal exposure to media was negatively associated with vaccination, although this association was not significant according to the final multivariate model. This negative association suggests that the media does not effectively convey public health messages, including information on the importance of basic childhood vaccinations. Furthermore, this lack of effectiveness may stem from low maternal literacy levels and insufficient community vaccination awareness. Hence, enriching media broadcasts with effective public health messages could be advantageous.

### Limitations

The SLDHS is a nationwide survey applying weighted analysis, thus making the results generalizable to the target population. Limitations of our study included the high percentage of missing values and the cross-sectional design of the secondary data analysis, making it difficult to establish a causal relationship. Additionally, we could not assess complete vaccination coverage (fully vaccinated) due to insufficient data regarding individual vaccines. Most of the information about childhood vaccination status was based on mothers’ or primary caregivers’ recall, and very little information was retrieved from vaccination documents; hence, recall bias cannot be ruled out.

## 5. Conclusions

Somaliland has low childhood vaccination coverage. Although the prevalence of those ever vaccinated throughout the country is low, great inequalities exist across regions. The inequitable allocation of healthcare resources and the national budget were associated with regional variation in childhood immunization coverage. Factors including the age of the children, maternal literacy, maternal number of ANC visits, and urbanicity of residence were positively associated with childhood vaccination coverage. Based on our results, expanding women’s educational programs to enhance maternal literacy, promoting awareness of timely vaccinations for children, and encouraging women’s ANC visits through various organizations may increase childhood vaccination coverage. Healthcare administrators should reassess the effectiveness of the national EPI, particularly in areas with low childhood vaccination rates. Specialized community outreach services, focusing on nomadic populations and community empowerment efforts involving collaboration with leaders, could increase participation in immunization campaigns. Implementing compulsory health cards for children during school enrollment may increase vaccination rates. Economically disadvantaged regions should receive increased budget allocations for healthcare coverage and immunization programs. Increased allocation of healthcare infrastructure is crucial for prioritizing regions with fewer healthcare facilities.

## Figures and Tables

**Table 1 vaccines-12-00509-t001:** Characteristics of the participants by vaccination status among children under five years (N = 2673).

Characteristics	Total	Vaccination Status		
	N = 2673	Never Vaccinated n (Weighted %) *	Ever Vaccinated n (Weighted %) *	*p*-Value
	N (%)	1763 (66%)	910 (34%)	
Age of the respondents (years)				
≤24	766 (28.7%)	486 (30.2)	240 (22.6)	0.001
25 to 34	1371 (51.3%)	803 (49.9)	617 (58)	
≥35	536 (20%)	319 (19.9)	207 (19.4)	
Maternal education level				
No education	2253 (84.3%)	1413 (87.8)	768 (72)	<0.001
Primary	348 (13%)	179 (11)	204 (19)	
Secondary and above	72 (2.7%)	16 (1)	91 (8.6)	
Ever attended school				
Yes	420 (15.7%)	196 (12.2)	295 (27.8)	<0.001
No	2253 (84.3%)	1413 (87.8)	768 (72)	
Maternal literacy status				
Cannot read at all	2142 (80.1%)	1375 (85)	684 (64)	<0.0001
Can read part of sentence	288 (10.8%)	125 (7.8)	170 (16)	
Can read whole sentence	243 (9.1%)	107 (6.7)	209 (20)	
Wealth index				
Poor	1628 (69.9%)	1081 (67.2)	371 (34.8)	<0.001
Middle	675 (25.3%)	377 (23.4)	329 (31)	
Rich	370 (13.8%)	151 (9.4)	364 (34.2)	
Maternal marital status				
Married	2545 (95.2%)	1538 (95.6)	1020 (96)	0.47
Divorced	78 (2.9%)	46 (3)	23 (2.2)	
Widowed	50 (1.9%)	24 (1.5)	20 (1.8)	
Employment status of the mother or caregiver				
Working	120 (4.5%)	56 (3.5)	86 (8)	<0.001
Not working	2553 (95.5%)	1553 (96.5)	976 (92)	
Father’s employment status				
Not working	1731 (64.7%)	1129 (70.2)	435 (40.9)	<0.001
Working	814 (30.5%)	409 (25.4)	586 (55)	
Not applicable ^1^	128 (4.8%)	71 (4.4)	43 (4)	
Maternal ANC visits				
No visit at all	2299 (86%)	936 (83)	239 (57)	<0.001
1 visit	104 (3.9%)	61 (5.4)	41 (9.8)	
2–3 visits	185 (6.9%)	92 (8.2)	93 (22)	
4+ visits	85 (3.2%)	39 (3.5)	46 (11.2)	
Parity				
1–2	654 (24.5%)	401 (25)	251 (23.6)	0.11
3–5	1203 (45%)	744 (46.3)	452 (42.5)	
6+	816 (30.5%)	464 (28.8)	360 (33.8)	
Age of the child in months				
0–23	1303 (48.7%)	846 (52.6)	337 (31.67)	<0.001
24–59	1370 (51.3%)	763 (47.4)	727 (68.33)	
Sex of the children				
Male	1382 (51.7%)	857 (53.3)	548 (51.5)	0.425
Female	1291 (48.3)	751 (46.7)	515 (48.5)	
Birth order of the child				
First baby	694 (26%)	412 (25.6)	325 (30.6)	<0.0001
Second baby	1198 (44.8%)	689 (42.8)	496 (46.6)	
Third baby or more	781 (29.2%)	508 (31.6)	242 (22.8)	
Place of delivery				
At home	2280 (85.3%)	1424 (88.5)	823 (77.3)	<0.001
Health facility	393 (14.7%)	185 (11.5)	241 (22.7)	
Number of children living at home				
1–2	763 (28.5%)	468 (29)	304 (28.6)	0.089
3–4	919 (34.4%)	563 (35)	324 (30.5)	
5+	991 (37.1%)	576 (36)	436 (41)	
Maternal media exposure				
Yes	289 (10.8%)	166 (10.3)	250 (24.4)	<0.001
No	2384 (89.2)	1443 (89.7)	804 (75.6)	
Place of residence				
Urban	289 (20.8%)	306 (19)	533 (50)	<0.001
Rural	726 (27.2%)	329 (20.5)	300 (28.2)	
Nomads	1390 (52%)	973 (60.5)	230 (21.7)	
Regions				
Awdal	216 (8.1%)	100 (6.2)	73 (6.8)	<0.001
Maroodijeex	194 (7.3%)	136 (8.5)	217 (20.4)	
Saahil	198 (7.4%)	74 (4.6)	39 (3.6)	
Togdheer	370 (13.8)	277 (17.2)	254 (24)	
Sool	837 (31.3%)	474 (29.5)	188 (17.7)	
Sanaag	858 (32.1%)	547 (34)	293 (27.5)	
Health care facility density				
Low ^1^	2263 (84.7%)	1373 (85.3)	774 (72.8)	<0.001
High ^2^	410 (15.3%)	236 (14.7)	280 (27.2)	
National budget allocation				
Low budgeting regions ^3^	1695 (63.4%)	1021 (63.5)	481 (45.2)	<0.001
High budgeting regions ^4^	978 (36.6%)	588 (36.5)	583 (54.8)	

* Data were weighted to represent the population. *p*-values were from Chi-squared test. ANC, antenatal care. ^1^ Regions with ≤ half of public healthcare facilities split by median values as the cutoff point. ^2^ Regions with > half of public healthcare facilities. ^3^ Regions with ≤ half of the national budget allocation split by median values as the cutoff point. ^4^ Regions with > half of the national budget allocation.

**Table 2 vaccines-12-00509-t002:** Multilevel logistic regression analysis of individual- and regional-level factors associated with childhood vaccination coverage among children under five years in Somaliland (N = 2673).

Characteristics (Variables)			Adjusted Odds Ratio (AOR) and 95% Confidence Interval (CI)		
	Model (I)	Model (II)	Model (III)	Model (IV)	Parsimonious Model (V)
Age of respondents (years)					
≤24		Ref	--	Ref	--
25–34		1.13 (0.83–1.54)	--	1.09 (0.80–1.49)	--
≥35		1.02 (0.74–1.40)	--	1.00 (0.74–1.37)	--
Ever attended school					
No	--	Ref	--	Ref	--
Yes	--	0.79 (0.51–1.23)	--	0.80 (0.50–1.28)	--
Women’s literacy status					
Cannot read at all	--	Ref	--	Ref	Ref
Can read part of sentence	--	1.71 (1.02–2.89) *	--	1.83 (1.08–3.07) *	1.61 (1.07–2.41) *
Can read whole sentence		2.16 (1.17–3.96) *		2.27 (1.24–4.15	1.89 (1.27–2.80) *
Wealth index					
Poor	--	Ref	--	Ref	--
Middle	--	0.60 (0.32–1.15)	--	0.67 (0.36–1.24)	--
Rich	--	1.25 (0.51–3.06)	--	1.35 (0.58–3.16)	--
Employment status of mother/caregiver					
Not working	--	Ref	--	Ref	--
Working	--	1.17 (0.60–2.26)	--	1.15 (0.60–2.20)	--
Father’s employment status					
Not working	--	Ref	--	Ref	--
Working	--	1.41 (0.99–2.99) *	--	1.30 (0.92–1.83)	--
Not applicable ^1^	--	1.02 (0.61–1.69)	--	1.09 (0.66–1.80)	--
Maternal ANC visits					
No ANC visit at all	--	Ref	--	Ref	Ref
1 ANC visit	--	2.02 (1.28–3.17) *	--	2.04 (1.31–3.17) *	2.06 (1.31–3.24) *
2–3 ANC visits	--	1.67 (1.08–2.58) *	--	1.85 (1.23–2.80) *	1.97 (1.34–2.90) **
4+ ANC visits	--	1.64 (0.96–2.78)	--	1.93 (1.14–3.26) *	2.08 (1.29–3.35) *
Birth order of the child					
First baby	--	Ref	--	Ref	--
Second baby	--	1.09 (0.86–1.39)	--	1.14 (0.90–1.44)	--
Third baby or above	--	0.89 (0.64–1.22)	--	0.94 (0.68–1.29)	--
Age of the child in months					
0–23 months old	--	Ref	--	Ref	Ref
24–59 months old	--	2.06 (1.58–2.69) **	--	1.94 (1.49–2.52) **	2.02 (1.58–2.58) **
Place of delivery					
At home	--	Ref	--	Ref	--
Health facility	--	1.35 (0.95–1.91)	--	1.30 (0.92–1.83)	--
Maternal media exposure					
No	--	Ref	--	Ref	--
Yes	--	0.61 (0.39–0.97) *	--	0.54 (0.35–0.83) *	--
Place of residence					
Nomads	--	Ref	--	Ref	Ref
Urban	--	8.02 (3.69–17.43) **	--	7.62 (3.63–16.0) **	7.45 (4.54–12.2) **
Rural	--	5.18 (1.81–14.75) *	--	4.55 (1.72–12.0) *	4.86 (2.30–10.2) **
Public health care facility density					
Low density ^2^	--	--	Ref	Ref	Ref
High density ^3^	--	--	2.15 (1.33–3.47) *	1.64 (1.03–2.63) *	1.64 (1.05–2.57) *
National budget allocation					
Low-budgeting regions ^4^	--	--	Ref	Ref	Ref
High-budgeting regions ^5^	--	--	1.52 (0.94–1.44)	1.54 (1.05–2.25) *	1.62 (1.11–2.36) *
ICC	0.288	0.091	0.296	0.090	0.099
PCV	Ref	0.691	−0.038	0.713	0.660
AIC	3179.903	2965.602	3160.852	2952.489	2974.877

* *p*-value < 0.05; ** *p*-value < 0.001; ANC, antenatal care; ICC, intraclass correlation coefficient; PCV, proportional change in variance; AIC, Akaike information criterion. ^1^ Widowed or no fathers were registered. ^2^ Regions with ≤ half of public healthcare facilities split by median values as the cutoff point. ^3^ Regions with > than half of public healthcare facilities. ^4^ Regions with ≤ half of the national budget allocation split by median values as the cutoff point. ^5^ Regions with > half of the national budget allocation. Ref refers to the reference group.

## Data Availability

Restrictions apply to the availability of these data. Data were obtained from Central Statistics, the Department of Ministry of National Planning, Somaliland, with the permission of Central Statistics, the Department of Ministry of National Planning, Somaliland.

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
