# Peer review of "Factors Associated with Vaccination Coverage among 0–59-Month-Old Children: A Multilevel Analysis of the 2020 Somaliland Demographic and Health Survey"

_vaccines, 2024, doi:10.3390/vaccines12050509_

Round 1

Reviewer 1 Report

Comments and Suggestions for Authors

This is a fascinating discussion examining factors associated with low childhood vaccination rates in Somaliland.  The results are not surprising, related to education and resources, and of course have been seen in the many other similar studies you cite.  Your suggestions for improvement are sound, funding and political will are needed to amend this problem. 

I have one question.  You do not mention a role of conflict.  I am not familiar with how recent conflict in the horn of Africa would affect this population.  Maybe the nomadic groups are able to steer clear of conflict.  But are battles and wars playing a role not only in availability of vaccines but also education of girls (leading to under educated mothers) and limited governmental support overall. 

Author Response

Thank you for your suggestions, comments, and questions. Indeed, East Africa, particularly the Horn of Africa, is a strategically significant and politically sensitive region, vulnerable to conflicts. However, it's worth noting that Somaliland has enjoyed political stability and peace for the past three decades, allowing for significant rebuilding efforts, including the healthcare system, following the collapse in 1991. While minor conflicts persist in some areas, the country as a whole is peaceful and politically stable. We added to the Discussion, “Somaliland has enjoyed political stability and peace for the past three decades and focused on rebuilding security, infrastructure, and healthcare systems, including the EPI program, over the past three decades.” (line 329-332) Regarding the absence of conflict-related factors in our article, it's because Somaliland, being mostly peaceful and politically democratic, has not been significantly affected by conflicts compared to neighboring countries like Somalia and Ethiopia. Nomadic groups may have managed to avoid conflict, and battles and wars have not played a significant role in limiting vaccine availability or education, particularly for girls. However, we appreciate your insight, and further exploration into this aspect could enrich our understanding.

Reviewer 2 Report

Comments and Suggestions for Authors

Major issue: it looks like the factors are compared one at a time. While I am not an expert in statistics, there must be a test/methods that studies/considers all factors at ones and identifies the most important ones

minor issues:

- Typo or mistake in "Almost four-fifths (86%) of mothers" . four-fifth is 80%, so almost four-fifths should mean just under 80%

- weighted percentages in Table 1 are confusing to me and should be explained    - what is the meaning of p-value in table 1? (p value for what test?)

Author Response

Thank you for your suggestion, comments, and questions.

Regarding the major issue, we conducted a multivariate analysis using logistic regression and compared five models to identify significant factors associated with the outcome (Table 2). The final model, the parsimonious model, was reported, which considers all factors simultaneously. Additionally, we utilized weighted analysis by incorporating women’s individual weights from the DHS in Table 1. The p-values reported in Table 1 represent the significance level of the chi-squared test. The purpose of using the weighted analysis is to balance discrepancies in the DHS data, since the nature of the DHS might be some areas are under-estimated while others are over-estimated during the sampling. we used weight to enhance the external validity of our study and reduce the bias.

We have addressed the minor issues you pointed out, including the clarification of percentages (line 233 and line 269 endnote to Table 1).

Reviewer 3 Report

Comments and Suggestions for Authors

The article covers an interesting topic and its aim is to assess factors associated with vaccine coverage in Somaliland, among children. The novelty is mainly related to the setting. Actually, there are few studies conducted among developing countries. Vaccine hesitancy and vaccine acceptance are highly influenced by socio-cultural aspects. therefore there is a need to contextualize and compare factors among countries.

The introduction is too long. I would suggest to focus on the most important background information. Some of the content could be moved to the discussion or simply removed.

the methods are well described and results are clearly reported. The tables are self-explicative and well-organized.

the discussion is coherent with the obtained results.

I would suggest adding a graèhical abstract to improve interest from the readers.

Author Response

Thank you for your suggestion, and comments.

We appreciate your recognition of the novelty of our study's setting and the importance of contextualizing and comparing factors among countries. We have shortened the Introduction from a word count of 1170 to 899 (line 57-106). We added a graphical abstract as per your comments.